

# Fracture-controlled fluid transport supports microbial methane-oxidizing communities at the Vestnesa Ridge

Haoyi Yao[1], Wei-Li Hong[1,2], Giuliana Panieri[1], Simone Sauer[1,2], Marta E. Torres[3], Moritz F. Lehmann[4], Friederike Gründger[1]and Helge Niemann[1,4,5]

[1]CAGE - Centre for Arctic Gas Hydrate, Environment and Climate, Department of Geosciences, UiT The Arctic University of Norway in Tromsø, Norway
[2]NGU-Geological Survey of Norway, Trondheim, Norway
[3]CEOAS- College of Earth, Ocean, and Atmospheric Sciences, Oregon State University, Corvallis, USA
[4]Department of Environmental Sciences, University of Basel, Switzerland
[5]NIOZ Royal Netherlands Institute for Sea Research, Department of Marine Microbiology and Biogeochemistry, and Utrecht University, The Netherlands

*Correspondence to*: Haoyi Yao (haoyi.yao@uit.no)

**Abstract.** We report on a rare observation of a mini-fracture in near-surface sediments (30 cm below the seafloor) visualized using rotational scanning X-ray of a core recovered from the Lomvi pockmark, Vestnesa Ridge west of Svalbard (1200 m water depth). Porewater geochemistry and lipid biomarker signatures revealed clear differences in the geochemical and biogeochemical regimes of this core compared with two additional ones recovered from pockmarks sites at Vestnesa Ridge, which we attribute to differential methane transport mechanisms. In the sediments core featuring the shallow mini-fracture at pockmark Lomvi, we observed high concentrations of both methane and sulfate throughout the core in tandem with moderately elevated values for total alkalinity, [13]C-depleted dissolved inorganic carbon (DIC), and [13]C-depleted lipid biomarkers (diagnostic for the slow-growing microbial communities mediating the anaerobic oxidation of methane with sulfate - AOM). In another core recovered from the same pockmark about 80 m away from the fractured core, we observed complete sulfate depletion in the top centimeters of the sediment and much more pronounced signatures of AOM than in the fractured core. Our data indicate a gas advection-dominated transport mode in both cores facilitating methane migration into sulfate-rich surface sediments. However, the more moderate expression of AOM signals suggest a rather recent onset of gas migration at the site of the fractured core, while the geochemical evidence for a well-established AOM community at the second coring site at the Lomvi pockmark suggest that gas migration has been going on for a longer period of time. A third core recovered from Lunde pockmark was dominated by diffusive transport with only weak geochemical and biogeochemical evidence for AOM. Our study highlights that advective fluid and gas transport supported by mini-fractures can be important in modulating methane dynamics in surface sediments.

## 1 Introduction

Large-scale fractures providing conduits for fluid/gas migration are commonly observed on seismic profiles (Tobin et al. 2001, Weinberger and Brown 2006, Plaza-Faverola et al. 2015). Increased sediment permeability due to fractures may facilitate gas and fluid transport. Indeed, macro-fractures were often observed in association with cold seep systems, where methane-rich





fluids from greater sediment depth reach shallow sediments or may even be transported across the sediment-water-interface (Berndt et al. 2014, Sahling et al. 2014). Prominent examples of fracture-controlled fluid migration at cold seep systems include locations such as Hydrate Ridge (Torres et al. 2002, Weinberger and Brown 2006, Briggs et al. 2011) and Blake Ridge (Egeberg and Dickens 1999). Seepage at these locations can sustain a tremendously high biomass of chemosynthetic

communities that either directly consume methane, or benefit from products of methane oxidation, such as sulfide (Boetius and Suess 2004, Niemann et al. 2013). Fractures visible on seismic profiles often exceed 10 m in length (Gabrielsen et al. 1998). However, surface sediments may also feature smaller-scale, branched fracture networks (hereafter referred to as mini-fractures) which propagate from macro-fractures as the fluid pressure increases (Friedman 1975, Briggs et al. 2011, Anders et al. 2014).

The role of small-scale fracture networks in routing methane upwards into the near-surface sediments is not well understood. In particular, the significance of mini-fractures for the development of methane-dependent microbial communities is poorly constrained. Geochemically, fractures facilitate migration of deep fluids that are laden with electron donors from deeper sediments, which can then be used by sedimentary microbes to maintain their metabolism. To date, such mini-fractures have

either been detected by X-ray images of cores under pressure (Riedel et al. 2006), or by the presence of macroscopic biofilms lining subseafloor fractures (Briggs et al. 2011). These biofilms were usually present at the sulfate-methane transition zone (SMTZ), where methane is consumed by a consortium of methanotrophic archaea and sulfate-reducing bacteria mediating the anaerobic oxidation of methane (AOM) with sulfate as the terminal electron acceptor (Knittel and Boetius 2009):

$CH_4 + SO_4^{2-} \rightarrow HCO_3^- + HS^- + H_2O$                                                                 Eq. (1)

In contrast to large scale transport pathways, mini-fractures are difficult to detect as they cannot be resolved with seismic tools (Emery and Myers 1996, Gabrielsen et al. 1998), and may play thus an underappreciated but potentially significant role in sediment methane dynamics, with regards to the efficiency of the benthic microbial methane filter.

In this study, we report on the presence of a mini-fracture in the near-surface sediments of the active pockmark Lomvi, located on the Vestnesa Ridge (79°N, 6°E), west of the Svalbard Archipelago. Using an interdisciplinary approach that combines geochemical and biogeochemical measurements, we investigate the effects that such mini-fractures may impose on the benthic methane dynamics and associated microbial communities. Our data show that the mini-fracture provided a conduit for

advective gas migration fostering AOM, but the moderate expression of AOM-associated biogeochemical signals suggest a rather recent opening of the fracture.

## 2 Material and Methods

### 2.1 Study sites

Vestnesa Ridge is NW-SE trending, ~100 km long and, covered with ~1km thick contourite drifted sediments. Vestnesa Ridge features numerous pockmark structures (see a more detailed description of the geological setting in Plaza-Faverola et al. (2015) and Panieri et al. (2017)). This ridge is part of a submarine gas-hydrate system on the west Svalbard margin (1200m water depth), where fluid and gas migration from deep hydrocarbon reservoirs towards the seafloor has potentially been going on

since the early Pleistocene (Knies et al. 2018). Past investigations have shown that the ridge actively releases methane gas





from the seafloor along the eastern segment of the structure (Bünz et al. 2012, Smith et al. 2014, Plaza-Faverola et al. 2015, Panieri et al. 2017), and seismic data suggest that seepage is related to intensive seabed faulting and rifting (Plaza-Faverola et al. 2015). The eastern part of Vestnesa Ridge features the pockmarks Lunde and Lomvi (Fig. 1), both belonging to the most active structures known in the area (Bünz et al. 2012, Panieri et al. 2017). Enhanced reflections and "push-down" features

observed in the seismic transects of Lunde and Lomvi were interpreted as chimney structures containing free gas, which originated from beneath the bottom of gas hydrate stability zone (Bünz et al. 2012, Smith et al. 2014). High excess pore pressure at the summit of this gas column fractured the sediments and led to the presence of free gas in the hydrate stability zone (Weinberger and Brown 2006, Bünz et al. 2012).

## 2.2 Sample collection

We investigated three sediment cores from Vestnesa Ridge: two were collected by a multicorer (MC) during cruise CAGE15-2 with R/V Helmer Hanssen (893MC and 886MC) in 2015, and an additional push core (008PC) was recovered during the P1606 cruise with R/V G.O.Sars and a remotely operated vehicle (ROV) Ægir about 80m away from 893MC in 2016. The

multicorer system can collect up to six cores per deployment in parallel, and a MISO (Multidisciplinary Instrumentation in Support of Oceanography- MISO, Woods Hole Oceanographic Institution) tow-cam was attached to the MC frame, allowing targeted video-controlled sampling (Panieri et al. 2015, 2017). Of the six cores, one was subsampled for porewater analyses, and two adjacent cores were used for lipid biomarker and headspace gas analyses, respectively. The cores for porewater extraction were used for x-ray analysis. 893MC was collected at the site with bacterial mats around outcropping carbonate

crusts at Lomvi pockmark, whereas 886MC was collected at a soft sediment site with patchy bacterial mats and tubeworms at the Lunde pockmark (Figs. 1 and 2). The push core 008PC was recovered from a large patch of bacterial mat at the Lomvi pockmark. This core was only sampled for the analysis of porewater and lipid biomarkers (limited sample material impeded the analysis of the gas composition from 008PC; Table 1).

Immediately upon recovery, cores were subsampled for biomarker, headspace and porewater analyses. Details on headspace

sampling and gas analysis in the multi cores were described in Panieri et al. (2017) and references therein. Porewater was extracted at ambient temperature (ca. -1 °C), at a resolution of 2 cm, with either 10cm (893MC and 886MC) or 5 cm rhizon samplers (008PC) (Seeberg-Elverfeldt et al. 2005) attached to acid-cleaned syringes. Rhizon membranes were soaked in Milli-Q water before use. The first half ml of the recovered porewater was discarded to avoid dilution/contamination with residual Milli-Q water in the rhizons. Sediment samples for biomarker analyses were collected with methanol-cleaned spatula onboard

at a resolution of 2 cm, wrapped in aluminum foil and subsequently stored frozen at -20 °C until analysis. Intact sediment cores were kept at 4 °C for further X-ray analysis in our home laboratories using a Geotek MSCL-XCT.

## 2.3 Porewater analyses

Total alkalinity (TA) was measured onboard by the Gran titration method a few hours after syringes were disconnected from the rhizon samplers. The HCl titrant (0.012M) was checked daily onboard with local surface seawater and 10mM borax to

ensure the quality of the acid. The pH meter of the titrator was calibrated with pH standard solutions (pH of 4, 7 and 11) both before and after the cruise. Details of the titration protocol can be found in Latour et al. (in review). Porewater aliquots (2 ml) for sulfate analysis were preserved with 3 mL zinc acetate solution (23M) to precipitate the dissolved sulfide (Gieske et al. 1991, Grasshoff et al. 1999) for CAGE 15-2 samples. All sulfate analyses were performed using a Dionex ICS-1100 Ion Chromatograph equipped with a Dionex IonPac AS23 column (Sauer et al. 2016). For sulfide concentration measurements,

the precipitated zinc sulfide was quantified in our home laboratories with a spectrophotometric method (Cline 1969) using a



UV-1280 UV-vis Spectrophotometer (Shimadzu). The amount of zinc acetate added to samples from core 008PC was too low to capture all the dissolved sulfide, thus the measured sulfide concentrations are minimum values. 2 mL aliquots of porewater were fixed onboard with saturated $HgCl_2$ (27mM final concentration) for the subsequent measurement of $\delta^{13}C$ of dissolved inorganic carbon (DIC) (Grasshoff et al. 1999). The $\delta^{13}C$-DIC of CAGE 15-2 samples were analyzed using a Finnigan DELTA-Plus mass spectrometer coupled to a Gas-Bench II as described in Torres et al. (2005). The $\delta^{13}C$-DIC in 008PC pore water was determined from the $CO_2$ liberated from the water after acidification with phosphoric acid. Measurements were done at EAWAG (The Swiss Federal Institute of Aquatic Science and Technology) using an IRMS (Isotope Ratio Mass Spectrometer, Isoprime) equipped with a Gilson 222XL Liquid Handler and a Multiflow unit (Isoprime). The standard deviation of the $\delta^{13}C$-DIC measurements from repeated measurements of a standard was ±0.1‰ (1σ, n=27). The stable carbon isotope values for DIC are reported in the conventional δ-notation in permille (‰) relative to V-PDB (Vienna Pee Dee Belemnite).

## 2.4 Lipid extraction, quantification, identification and determination of compound-specific stable carbon isotope composition.

Lipid biomarkers were extracted and further analyzed according to previously reported protocols (Elvert et al. 2003, Niemann et al. 2005, Blees et al. 2014). Briefly, a total lipid extract (TLE) was obtained by ultrasonication of ~20g wet sediment samples in four extraction steps with solvents of decreasing polarity: dichloromethane (DCM)/methanol (MeOH) 1: 2; DCM/ MeOH 2: 1; and DCM for the last two extraction steps. The TLE was then saponified, and a neutral lipid fraction was extracted prior to methylation of the remaining polar fraction (comprising free fatty acids) to yield fatty acid methyl esters (FAME) for chromatographic analysis. Double bond positions of FAMES were determined by analyzing dimethyl-disulfide adducts (Nichols et al. 1986; Moss and Lambert-Fair 1989). The neutral fraction was further separated into hydrocarbons, ketones, and alcohols, the latter of which was derivatized to form trimethylsilyl adducts for analysis.

Individual lipids molecules were analyzed using a gas chromatograph (GC; Thermo Scientific TRACE™ Ultra), equipped with a capillary column (Rxi-5ms, 50 m, 0.2 mm id, 0.33 μm df), using helium gas as a carrier gas at a constant flow of 1 mL min$^{-1}$. The initial oven temperature was set to 50 °C, held for 2 min and then increased to 140 °C at a rate of 10 °C min$^{-1}$, held for 1 min, then further increased to 300°C at 4 °C min$^{-1}$. The final hold time was 63 min to analyze FAMEs or 160 min to analyze larger (i.e., high boiling point) lipids in the hydrocarbon and alcohol fractions. Concentrations were determined by flame-ionization detection (FID) using internal standards. Unknown compounds were identified with a quadrupole mass spectrometry unit at the chromatography periphery (Thermo Scientific DSQ II, Blees et al. 2014). Similarly, compound-specific stable carbon isotope ratios were determined using a magnetic sector isotope ratio mass spectrometry unit (Thermo Scientific Delta V Advantage) coupled to a gas chromatography setup with the above-outlined specification (Blees et al. 2014). $\delta^{13}C$ values are reported with an analytical error of ± 1‰.

## 3 Results and Discussion

### 3.1 Sediment X-ray imaging and porewater geochemistry

Our detailed X-ray imaging of cores retrieved from locations of known methane seepage in the Vestnesa Ridge revealed a mini-fracture in the Lomvi core 893MC (Fig. 3). Upon retrieval, this core showed extensive gas ebullition. It is possible that the fracture expanded during core retrieval because of pressure-induced volume changes of sedimentary gases. Nevertheless, methane analyses revealed a significant increase in methane concentration along the observed fracture (Fig. 4). Typically, high sediment methane concentrations in marine sediments lead to elevated rates of AOM, which in return lead to sulfate depletion



and sulfide production, and thus the development of a SMTZ. Furthermore, excess production of DIC during AOM leads to elevated sediment TA and low $\delta^{13}$C-DIC values. The marked methane increase at coring site 893MC was not paralleled by changes in other parameters, which are commonly associated with AOM. Rather, the smooth porewater profiles of sulfate, sulfide, TA and $\delta^{13}$C of DIC in this core seem typical for locations with low methane input, as often found in settings

characterised by diffusive transport regimes (Fig. 4a) (Treude et al. 2003, Niemann et al. 2009, Egger et al. 2018). We attribute this apparently contradictory observation of enhanced methane concentrations on the one hand and the rather 'inconspicuous' values of geochemical and biogeochemical parameters on the other to a recent genesis of the fracture (see further discussion below). To further investigate the changes associated with the highly heterogeneous nature of methane dynamics in this region, we compared this observation with two additional cores from contrasting settings in the Vestnesa Ridge.

A push core (008PC) retrieved from an active venting site (ca. 80 m to the SE of core 893MC) within the Lomvi pockmark showed sulfate-depletion within the first 5 cmbsf (Fig. 4b), indicating a high methane flux and a shallow SMTZ (Reeburgh 2007). This shallow SMTZ is comparable to those typically observed at locations of high methane flux, such as the Beggiatoa fields at Hydrate Ridge (Treude et al. 2003), the Gulf of Mexico (Ussler and Paull 2008) or Haakon Mosby Mud Volcano

(Niemann et al. 2006a). At these high flux sites, AOM rates have been estimated to be in the order of several mmol m$^{-2}$d$^{-1}$. A third core (886MC) was retrieved from a soft-sediment site covered with tubeworms and small patchy bacterial mats spots (Fig. 2) in the adjacent active Lunde Pockmark. Sulfate concentrations in this core showed only a moderate decrease with sediment depth and no methane was detected in the upper 20 cm of the core (Fig. 4c). These data, are consistent with observations of low sulfide concentrations and TA. Together, these data indicate a significantly lower methane flux and

efficient methane-carbon retention by AOM in the sediments at this coring site (Niemann et al. 2006a, b, Levin et al. 2016). Although Lunde core 886MC is located in a diffusive system, the convex shape of the sulfate concentration profile along with increasing methane concentration at the bottom suggest non-steady state conditions. The convex shape of the sulfate profile can be related to an on going increase in methane flux (Fischer et al. 2013, Hong et al. 2017a). It may also be related to the intrusion of seawater into the shallower sediments, which can be induced by bioventilation, and/or ascending methane bubbles

from the sub-seafloor (Haeckel et al. 2007, Hong et al. 2016). Our visual investigations of the seafloor revealed the presence of tubeworms but other fauna typically associated with bio-irrigation, such as bivalves were not present. Methane concentration in the upper sediment section was very low, and we did not observe methane bubbles emanating from the seafloor at the coring site. We thus assume that moderate bio-irrigation and a recent increase of the diffusive methane flux at the coring site can explain the non-steady state sulfate and methane profiles in the Lunde pockmark core.


### 3.2 Methanotrophic community development

To further investigate the role of the detected mini-fracture in core 893MC on the ambient biogeochemistry and microbial community, we investigated archaeal and bacterial lipid biomarkers and their associated stable carbon isotope signatures that are diagnostic for AOM communities (Niemann and Elvert 2008) and references therein. ANME typically produce a suite of

glycerol ether lipids comprising isoprenoidal alkyl moieties that may also occur as free hydrocarbons in environmental samples. We found the isoprenoidal dialkyl glycerol diethers archaeol and *sn-2*-hydroxylarchaeol in all three cores (Fig. 4a-c). Furthermore, the $^{13}$C-depleted signatures of these compounds provide evidence that their source organisms mediate sulfate-dependent AOM. Indeed, ANME biomass is characterized by a strongly $^{13}$C-depleted isotope composition because the metabolized methane is typically $^{13}$C-depleted, and AOM is associated with a strong kinetic isotope effect (Whiticar 1999).

The sulfate-reducing partner bacteria in AOM produce characteristic fatty acids (e.g., C16:1$\omega$5c, and cyC17:0$\omega$5,6) which we observed at relatively high concentrations (Fig. 4 a-c). As these bacteria incorporate $^{13}$C-depleted DIC produced by the anaerobic methanotrophs (Wegener et al. 2008), their stable carbon isotope signature was also depleted in $^{13}$C. Our biomarker




data are consistent with an active AOM microbial population at all Vestnesa Ridge sites.

Our data also show, however, clear differences in the abundance of AOM-derived lipids at the three investigated coring sites (Figs. 4a-c). To highlight these differences, we calculated average concentrations and $^{13}C/^{12}C$ isotope ratios of archaeol and the fatty acid C16:1ω5c (i.e. typical ANME and associated SRB lipids) from the three cores and compared these values to a non-seeping reference site from south of Svalbard (Hong et al. 2017a, Yao et al. 2017) and a known high flux site at Hydrate Ridge (Elvert et al. 2005) (Fig. 5). We found the lowest concentrations of the diagnostic lipids at the non-seeping reference site, followed by the Lunde core 886MC, the Lomvi core 893MC then 008PC and finally the Hydrate Ridge core. The significantly higher concentrations of AOM-derived lipids at the Lomvi (particular in core 008 PC) compared to the Lunde site (core 886MC) is consistent with the geochemical signals of AOM (e.g. sulfate, sulfide, $\delta^{13}C$ of DIC) in the respective cores. The differences in concentrations of diagnostic lipids suggest a high standing stock of AOM communities in core 008PC, and a much lower one in the other two cores. AOM communities grow very slowly, with doubling times of several months (Nauhaus et al. 2007, Zhang et al. 2011, Timmers et al. 2015). A sudden increase in methane concentration in the sulfate-rich sediments comprising only a small initial standing stock of AOM microorganisms, may eventually lead to elevated AOM activity, but with a significant lag time of several months to years. Our biomarker data thus indicates that the increase in methane concentrations at coring site 893MC occurred rather recently, with too little time for the development of a more mature AOM community as for example observed at the site 008PC at the same pockmark or at another mini-fracture at Hydrate Ridge (Briggs et al. 2011).

Because of the spatial dynamics of venting at this location (Bohrmann et al. 2017, Hong et al. 2017b), it is likely that the biomarker results reflect the cumulative history of microbial AOM activity, rather than solely the actual situation. Nonetheless, we observed a general decrease in $\delta^{13}C$ of both bacterial and archaeal lipids in horizons of present day sulfate depletion indicating a higher contribution of AOM-derived compounds to the lipid pool. Such decrease in $\delta^{13}C$ was apparent at ~10 cmbsf in 893MC where sulfide started to accumulate, at ~5 cmbsf in core 008PC where sulfate was depleted, and at 10-15 cmbsf in core 886MC where methane began to increase downcore (Fig. 4). At these depths, the ratios of $sn$-2-hydroxylarchaeol to archaeol were 0.98 (Lomvi core 893MC), 0.37 (Lomvi core 008PC) and 0.26 (Lunde core 886MC), indicating ANME-1 as a likely candidate mediating AOM at all the investigated coring sites (Niemann and Elvert 2008). The known SRB types associated with ANME-1 and ANME-2 belong to the *Deltaproteobacteria, Desulfosarcina/Desulfococcus* clade (Seep-SRB1), which typically display distinct ratios of the fatty acids C16:1ω5c relative to isoC15:0. In systems dominated by Seep-SRB1 associated to ANME-1 this value is commonly <2, while it is >>2 in systems where Seep-SRB-1 is associated with ANME-2 (Niemann and Elvert 2008). At all coring locations, this biomarker ratio was >2 (3.2, Lomvi cores 893MC; 5.4, Lomvi core 008PC; 7.9, Lunde core 886MC), which is indicative of an SRB eco type associated with ANME-2 rather than with ANME-1. We can only speculate about these contradicting lipid patterns and would need additional DNA-based tools to further identify the AOM key microbes at the investigated sites.

The $^{13}C$-values of ANME-associated SRB lipids is mainly influenced by the $\delta^{13}C$ -value of source methane-carbon (Summons et al. 1994, Riou et al. 2010), though other environmental parameters such as substrate availability and temperature are also known to influence lipid $\delta^{13}C$-signals. The reference site showed $\delta^{13}C$ values of archaeal and bacterial lipids that were not conspicuously depleted. This indicates a low/negligible standing stock of AOM communities at the site. Here, archaeal and bacterial lipids are likely originated from processes other than AOM (e.g. organic matter degradation by heterotrophs). At site 893MC, the $\delta^{13}C$ values of archaeal and bacterial lipids were not as depleted as at the other three sites. The difference in $\delta^{13}C$ signature of archaeol with respect to the source methane ($\delta^{13}C$-CH$_4$ = -57.8 ‰ in 893MC and -62.9 ‰ in 886MC, data from Panieri et al. (2017)), $\Delta\delta^{13}C$, in 893MC were slightly lower than in 886MC (Fig. 5a). This may reflect an overprint of lipids





derived from organisms mediating non-AOM process (e.g. organic matter degradation) by those derived from AOM, which supports our assumption that the mini-fracturein 893MC and the associated AOM community developed rather recently. Assuming a uniform source methane value of XX for the Lomvi pockmark, site 008PC showed the highest $\Delta\delta^{13}C$ values (Fig. X). Together with the high standing stock of AOM communities and the rapid depletion of sulfate in this core, this indicates the highest AOM activity at site 008PC among the three investigated cores. Although the standing stock of AOM communities was seeming lower at site 886MC, the $\Delta\delta^{13}C$ values were similar when compare to 008PC. This suggest that AOM-communities also dominated the microbial community despite the lower methane flux at site 886MC.

**Summary and conclusion**

At the coring site at the Lunde pockmark, methane transport is dominated by diffusion, while at the Lomvi pockmark, we found evidence for advective methane transport, with indication for different onsets of gas seepage at the different coring sites (Fig. 6). Together with the porewater geochemical constrains, the distribution of $^{13}C$-depleted lipid biomarker underscores that the pockmark methane biogeochemistry is differentially affected by the advective vs. diffusive transport regimes in these three cores. In sediments where methane availability is limited by the diffusive supply, only a comparably low standing stock of AOM community developed. Our data demonstrated that mini-fractures (as observed in 893MC) represent important pathways for methane, facilitating the development of an active AOM community, and yet a high standing stock of the slow-growing AOM communities require that advective transport to proceed for an extended period of time. Mini-fractures are rarely recognized because they are below the resolution of seismic tools and their detection is mostly incidental. Yet our study clearly highlights their relevance for benthic methane dynamics and adds to the very limited knowledge on the fracture-networks contribution to carbon cycling.

**Data availability**

All the data in the manuscript can be found in the supplementary.

**Author contributions**

H.Y. and G.P. collected biomarker samples, W-L. H., M.E.T., and S.S. contributed to porewater sampling and analyses. H.N. and M.F.L. supported the overall lipid biomarker analyses. H.Y. wrote the majority of the manuscript. G.P and H.N. supervised the research. All authors contributed to the discussion of data and the writing of the paper at different stages.

**Competing interests**

We declare no conflict of competing interest for the co-authors.



**Acknowledgements**

We would like to acknowledge the captains, crews and all scientists on board R/V Helmer Hanssen for cruise CAGE 15-2 and R/V G.O. Sars for cruise P1606. We appreciate the team from Woods Hole Oceanographic Institution (WHOI) MISO (Multidisciplinary Instrumentation in Support of Oceanography) for the Towcam system and the ROV Ægir pilots. We would further like to thank Dr. Carsten J. Schubert and Serge Robert from EAWAG for the support during carbon isotopic analyses. This work was supported by the Research Council of Norway through its Centres of Excellence funding scheme (project number 223259). The publication charges for this article have been funded by a grant from the publication fund of UiT The Arctic University of Norway.

Table 1: Coring location, seafloor habitat information, and performed analyses. Abbreviations: Bac. mats, bacterial mats.; Carb, carbonate crust; DIC, dissolved inorganic carbon.

Figure 1. Regional multibeam bathymetric map of Vestnesa Ridge showing the Lunde and Lomvi pockmarks and sampling locations. Locations of multicores and push cores used in this paper (red stars).

Figure 2. Still images of the TowCam deep-sea imaging system before launching of the multicore (a. 893MC and c. 886MC), and ROV guided push core (b. 008PC). 893MC (a) shows network structure bacterial mats (green laser points are 20cm apart); 008PC (b) shows a large patch of white bacterial mats and black sediments (the core diameter is 8.5cm); 886MC (c) shows soft sediments with tubeworms and small patchy bacterial mats (green laser points are 20cm apart).

Figure 3. X-Ray images of 893MC (a). The different rotational planes show an X-Ray transparency extending throughout core 893. This zone is interpreted as a zone of weakness facilitating fluid and gas migration in situ. The void probably became gas filled after core recovery. Rotational video of this core is available in the supplementary materials.

25

Figure 4. Biogeochemistry profiles. First row: Methane (black dots), archaeal lipid concentrations (μg/g of dry sediment) (dots), and $\delta^{13}C$ values (in ‰ VPDB) (triangles). Second row: Sulfate (black dots), bacterial lipids concentrations (μg/g of dry sediment) (dots), and $\delta^{13}C$ values (in ‰ VPDB) (triangles). Third row: Total alkalinity (blue triangles), $\delta^{13}C$ of dissolved inorganic carbon (DIC)(in ‰ VPDB) (red diamonds) and sulfide concentration (maroon cross) in the three cores 893MC (left), 008PC (middle) and 886MC (right).

Figure 5. Summary of lipid biomarker data from a non-seeping site south of Svalbard (1522GC, 76.107°N, 15.957 °E, averaged from 0-350cm) (Yao et al. 2017), 893MC (averaged from 0-35cm), 008PC (averaged from 0-33cm), 886MC (averaged from 0-38cm) and a high flux site at Hydrate Ridge (Elvert et al. 2005) (averaged from 0-19cm). The $\Delta\delta^{13}C$ of archaeol relative to source methane in (a), and archaeol concentration in (b). The $\Delta\delta^{13}C$ of fatty acid C16:1ω5c in (c) and concentration in (d). The





values in a non-seeping site and at the three investigated cores were averaged for the whole core, whereas the reported values in Elvert et al., 2005 were only in the high activity section of the core.

Figure 6. Schematic illustration of different methane transport modes in the study area. Low amounts of methane in a diffusion dominated setting sustain a weakly defined AOM microbial community (right), mini-fracturing enhances the methane availability and triggers AOM microbial community growth (left), AOM community is fully developed as advective methane transport continues for a longer period of time, e.g., via lateral intrusion through a mini-fracture (middle).

Table 1.

| Pockmark (cruise) | Lomvi (CAGE15-2) | Lomvi (P1606) | Lunde (CAGE15-2) |
|---|---|---|---|
| Core | 893MC | 008PC | 886MC |
| Coordinates | 79° 0.180′ N | 79° 0.162′ N | 79° 0.366′ N |
|  | 6° 55.434′ E | 6° 55.488′ E | 6° 54.030′ E |
| Habitat | bac. mats & carb. | bac. mats | tubeworms |
| Methane headspace | × | N. A. | × |
| Porewater analyses | Sulfate, sulfide, TA, $\delta^{13}$C-DIC | Sulfate, sulfide, TA, $\delta^{13}$C-DIC | Sulfate, sulfide, TA, $\delta^{13}$C-DIC |
| Lipid biomarkers | × | × | × |

TA: total alkalinity

20

25

Fig. 1.





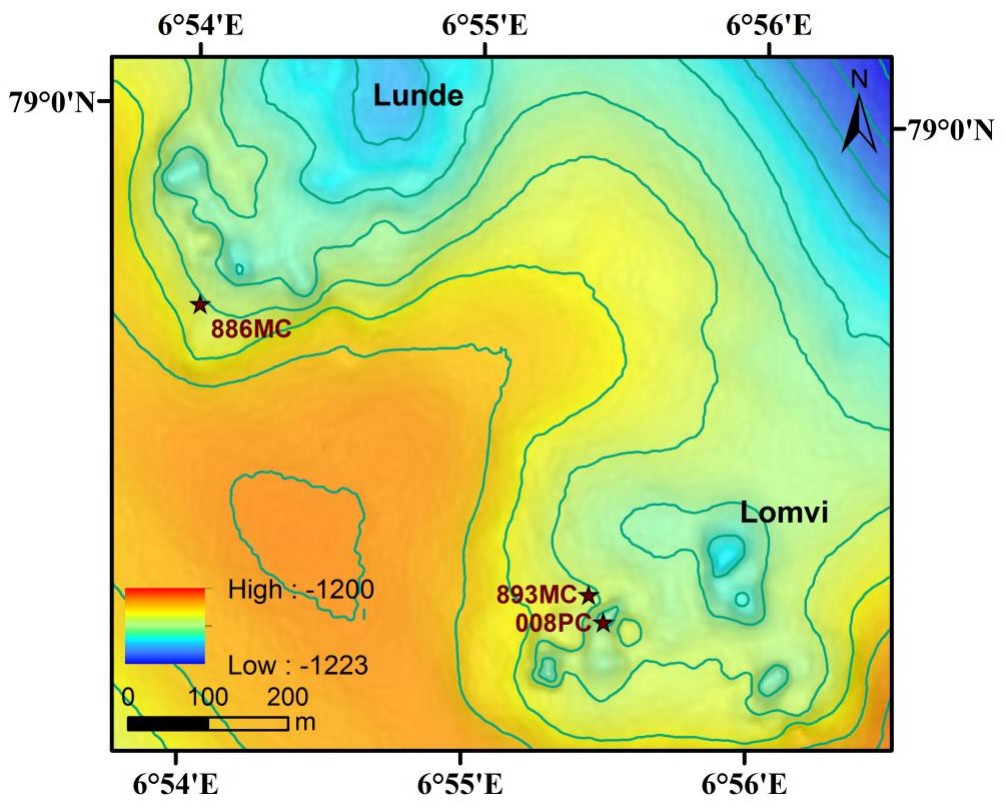

Fig. 2.

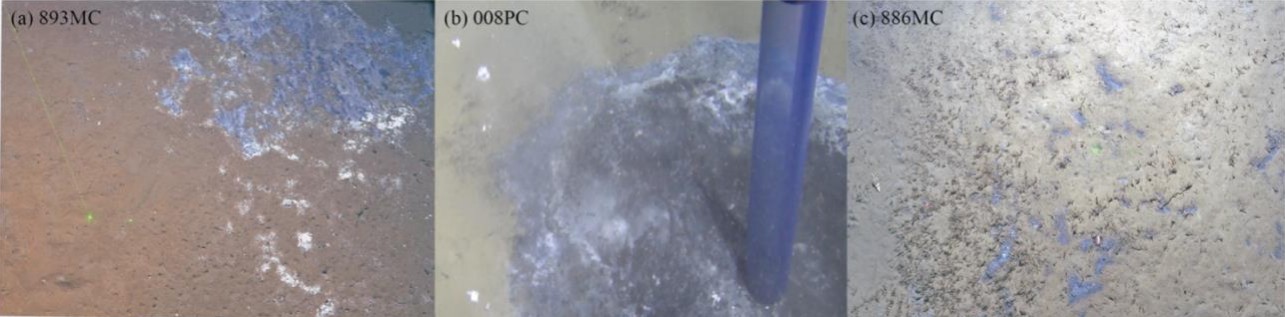





Fig. 3.

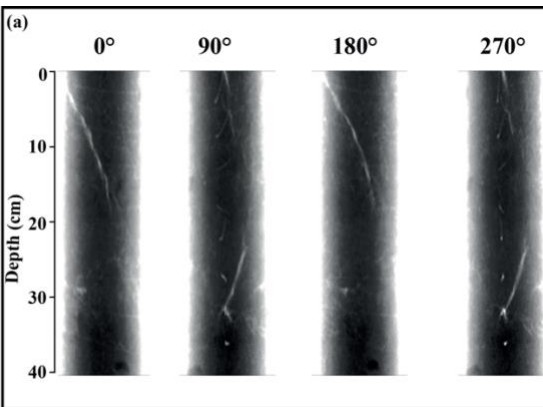

20

25  Fig. 4.



Fig. 5.



Fig. 6.



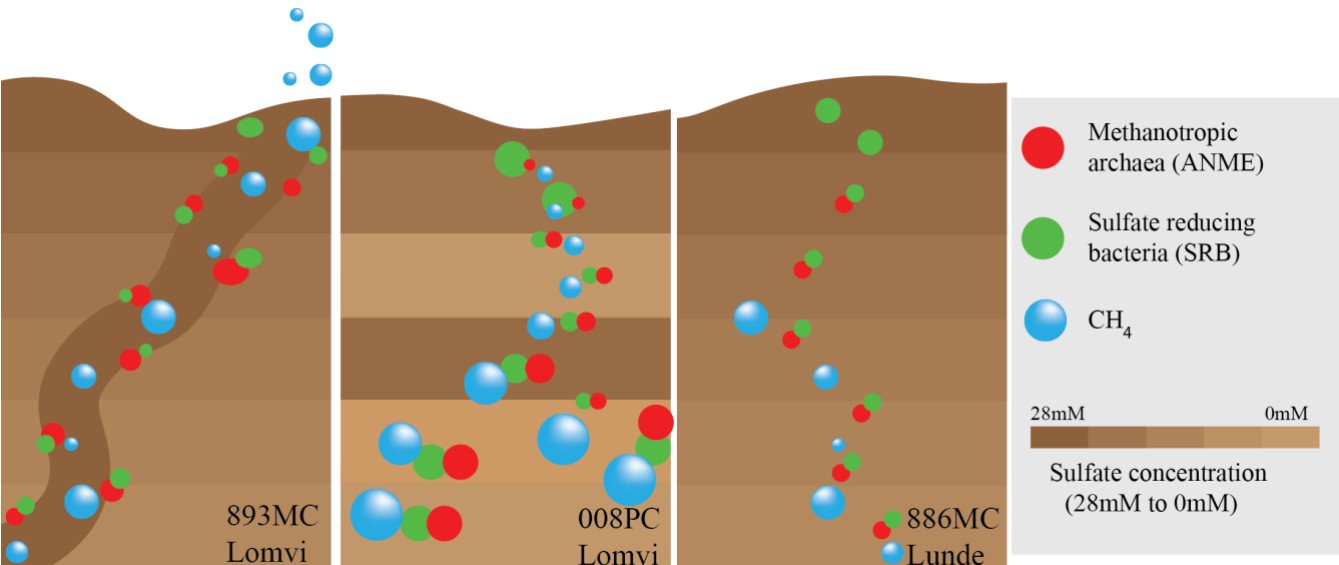

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
