# Peer review of "Fracture-controlled fluid transport supports microbial methaneoxidizing communities at the Vestnesa Ridge"

_Biogeosciences, 2018_

## Referee Comment (RC1) · Anonymous Referee #1 · 15 Sep 2018

In this manuscript the authors reveal the geomicrobiology of a fracture in the top 10s-cms sediment of the Vestnesa Ridge in comparison to that of two other un-fractured sediment horizons. They use pore fluid and gas phase geochemistry, and lipid biomarker, data to elucidate the geomicrobial processes of these sediments. The subject matter and findings of this study would be fully suitable for publication after the following few concerns are addressed. 1. The authors have used multicorer and push-coring techniques to retrieve the sediment samples. The fractured core (893 MC) in particular was retrieved by the multicorer. While it remains uncertain as to how faithfully a 40 cm core retrieved by this process can preserve small-scale fracture networks and present them for analysis, the authors too seem to have appreciated the

possibility that high internal gas pressure released at the time of the exposure of the core to atmospheric pressure (which is far low than its in situ pressure) may have expanded the fracture. The most remarkable fact, also according to the authors, is that despite the above issues, methane significantly increases in concentration as we go up the fracture from the bottom (Fig. 4). My concern is that the text offers no clear explanation regarding why methane concentration is higher near the sediment surface than at the bottom. Whilst a robust explanation of this is absolutely necessary, I would additionally recommend that the lipid biomarker study be complemented with metagenomic and/or metatrancriptomic analyses to reveal the microbial communities more objectively. 2. For smoother reading and comprehension of the present manuscript, the authors should coin such identifiers (or names) for individual sediment cores that harness both the core numbers as well as the coring-site names. 3. In the Methods section, the procedure for measuring methane concentration (the central parameter of this study) must be described and not entrusted entirely on cross referencing. 4. The labels of Figure 2 and 4 lack resolution. 5. Page 1 Line 22: I think the readers will find it easier to navigate the data if you say here "In a separate un-fractured core, recovered from the same pockmark approximately 80 m away from the fractured core, we observed . . . . . . . . . . .". 6. Page 1 Line 34: "Increased sediment permeability due to fractures may facilitate gas and fluid transport." – I think one should also consider seepage of sulfate, nitrate, etc. from the water into the sediment. Next sentence: no need to start with "Indeed". 7. Page 2 Line 4: Words like "tremendously" are subjective and judgmental so should be avoided. 8. Page 2 Line 5: Instead of obscure words such as "consume methane", or "benefit from . . . . . . sulfide" one should use specific terms as "oxidize". 9. Page 2 Line 13: What does "poorly constrained" mean here? In the context of the sentence, how can the significance of something be poorly constrained? 10. Page 2 Line 14: "maintain their metabolism" is technically not a right phrase; better say, ". . . . . . can then be used by sedimentary microbes as metabolic / bioenergetic substrates". 11. Page 2 Line 23: "thus" or "therefore" should come before "play". 12. Page 2 Lines 27-28: "Using an interdisciplinary approach that combines

geochemical and biogeochemical measurements, we investigate the . . ...." – There can be nothing like "biogeochemical measurements"; the whole thing is biogeochemical, so geochemical and biogeochemical are two redundant words; moreover lipid profile is a microbiological issue, and when combined with isotope ratios the issue becomes geomicrobiological. So this phrase should be written as, ". . . . . . . . geochemical and geomicrobiological methods . . . . . .." 13. Page 2 Line 37: Must introduce the reader to what "pockmark structures" are; can't pass on the onus of explaining this key concept to other citations. 14. Page 3 Line 4: 'Enhanced reflections and "push-down" features . . . . ...' – Please explain these terminologies so as to make the paper more comprehendible and attractive in its professed interdisciplinary context. 15. Page 3 Line 6: "High excess" – it's a redundancy of words; just say "Excessive pore pressure". 16. Page 3 Line 31: ". . . . . ..X-ray analysis in our home laboratories" – I think the authors meant "in the on land laboratory (rather than in the on board or on ship laboratory)"; so it should be written that way. Otherwise it seems that the authors had installed the Geotek MSCL-XCT in their homes. 17. Page 4 Line 36: "Nevertheless, methane analyses revealed a significant increase in methane concentration along the observed fracture (Fig. 4)." – When you just say "along the observed fracture" and don't mention the trajectory (whether from top to the bottom or upward from the bottom) then the reader gets to construe the first. But actually from top to the bottom of 893 MC, methane is not increasing rather decreasing. So please mention the specific trajectory. 18. Page 7 Lines 3-4: 5. "Assuming a uniform source methane value of XX for the Lomvi pockmark, site 008PC showed the highest $\Delta\delta$13C values (Fig. X)." – major oversight behind this incomplete statement – please rectify (what would be the value of XX?). 19. Table 1 should have a one-line description in relation what is listed in it; for instance, location of coring-stations, seafloor habitat information, analyses performed for individual sample-sites, etc. Incomplete / abbreviated expressions, such as "bac. mats & carb" for crucial habitat-related information, in Table 1, should be avoided. In Table 1 it is appearing that methane concentration and lipid biomarker analysis were not determined in the whole study although that was actually not the case; please

amend.

---

## Referee Comment (RC2) · Anonymous Referee #2 · 27 Nov 2018

The manuscript entitled "Fracture-controlled fluid transport supports microbial methane-oxidizing communities at the Vestnesa Ridge" by Yao et al deals with the geochemical and geomicrobiological aspects with respect to methane, of the cores from different sites at the Vestnesa Ridge. Also comparative studies have been made between the present study sites and that of non-seeping reference site and a high flux Hydrate ridge. The main emphasis was laid on the mini -fracture observed in the near surface sediments. However, the following points need to be relooked at

1.Clarity needs to be brought into the naming of the cores, their depth, site distance from each other.  It is unclear when the non-seeping reference site and a high flux

Hydrate ridge site is used in the manuscript for comparison. It is unclear whether only the Lomvi pockmark core has a min fracture or the other cores also had and to what extent. Also when a comparison is done between cores with respect to the fracture zone its important to know how intense was the fracture in all the cores compared and then that can be discussed with respect to the microbial community therein.

2. Please explain the precaution taken during coring so that the reliability of the extent and presence of a mini-fracture is confirmed.

3. Also there are contradictory findings in this manuscript which needs to be justified appropriately rather than just be assumed.

3. Pg 3 Line 31 & Line 40 Repalce the word 'home' with the name of the laboratory

4. Pg 3 ln 36 In the methodology the statement 'Details of the titration protocol can be found in Latour et al. (in review)' is not reader friendly as the paper is under review so it would better to specify the method used.

5. Pg4 Ln 13-14 there are three references sited which part of the protocol has been taken from which reference is not clear to the reader, either it should be given clearly or details should be elaborated in the methodology section.

5. Pg 5 Line 5-7 Needs reframing to bring out clarity to the reader

6. Pg 4 Line 5 What does the word 'highest' mean, it can range from any number, please specify quantitatively

7. Pg 4 ln 37. It is not clear how methane concentrations were determined from the fracture zone.

8. Pg 5 ln5-7 Though the contradictory observations are attributed to the recent development of the fracture, it is not clear as to what could be the time period for the word 'recent'.

9. Pg 5 ln 28 The authors are assuming a process. It would be better if the authors only

explain the possible conditions or mention the factor that could lead to such a function.

10. pg 6 ln 20 pls specify the location

---

## Author Comment (AC1) · 6 Dec 2018

Authors reply to Referee #1:

We thank the anonymous reviewer's positive comment and suggestion. Below we reply to the comment and suggestions one by one.

Referee's comment (RC): 1. The authors have used multicorer and push-coring techniques to retrieve the sediment samples. The fractured core (893 MC) in particular was retrieved by the multicorer. While it remains uncertain as to how faithfully a 40 cm core retrieved by this process can preserve small-scale fracture net- works and present

them for analysis, the authors too seem to have appreciated the possibility that high internal gas pressure released at the time of the exposure of the core to atmospheric pressure (which is far low than its in situ pressure) may have ex- panded the fracture. The most remarkable fact, also according to the authors, is that despite the above issues, methane significantly increases in concentration as we go up the fracture from the bottom (Fig. 4). My concern is that the text offers no clear explanation regarding why methane concentration is higher near the sediment surface than at the bottom.

Authors's reply (AR): We thank the reviewer for suggesting to clarify the concentration differences in the fractured core. We will edit the text accordingly offering the explanation that the methane concentration is higher near the sediment surface than at the bottom. Indeed, the fact that the methane concentration at the top sections of the core were higher is a very remarkable fact and we will highlight this further in the revised version of the manuscript. Obviously, the pressure change during core retrieval will have led to bubble formation. However, although outgassing effects may have led to a smearing of the methane concentration profile, the pressure change will have affected both the top and bottom of the core equally (both of which contained CH4 concentrations above saturation level at atmospheric pressure). We thus argue that already in situ, methane concentrations were higher at the top of the core, possibly because we didn't fully capture the fracture but only the upper part at the sediment surface (as the reviewer rightly points out). Indeed, our X-ray analysis suggest the prevalence of the fracture in the upper 30 cm of the sediment. We will better explain this in the revised version of the MS.

RC: Whilst a robust explanation of this is absolutely necessary, I would additionally recommend that the lipid biomarker study be complemented with metagenomic and/or metatrancriptomic analyses to reveal the microbial communities more objectively.

AR: Thanks for the reviewer's recommendation for adding the metagenomic or meta-transcriptomic analyses. While the reviewer is right that such analyses would further add to the ID and functioning of the microbial community, we have to admit that this is

beyond the author's PhD project, so that we would like to refrain from running further analyses.

RC-2. For smoother reading and comprehension of the present manuscript, the authors should coin such identifiers (or names) for individual sediment cores that harness both the core numbers as well as the coring-site names.

AR: We thank the referee's suggestion and will change the text accordingly.

RC-3. In the Methods section, the procedure for measuring methane concentration (the central parameter of this study) must be described and not entrusted entirely on cross referencing.

AR: We will update the method section with a brief description of the method for methane measurement.

RC-4. The labels of Figure 2 and 4 lack resolution.

AR: We will change the resolution of the labels in Figure 2 and 4.

RC-5. Page 1 Line 22: I think the readers will find it easier to navigate the data if you say here "In a separate un-fractured core, recovered from the same pockmark approximately 80 m away from the fractured core, we observed . . .. . .. . .. . .".

AR: We will change the description here for a clear navigation of the geographic position of these cores.

RC-6. Page 1 Line 34: "Increased sediment permeability due to fractures may facilitate gas and fluid transport." – I think one should also consider seepage of sulfate, nitrate, etc. from the water into the sediment. Next sentence: no need to start with "Indeed".

AR: Thanks for the reviewer's comment, we have discussed a little on the water column solutes such as sulfate are drawn into the sediment, but we will make this more clear in the revised version.

RC-7. Page 2 Line 4: Words like "tremendously" are subjective and judgmental so should be avoided.

AR: We will change the wording accordingly.

RC-8. Page 2 Line 5: Instead of obscure words such as "consume methane", or "benefit from ...... sulfide" one should use specific terms as "oxidize".

AR: We will changed the wording here.

RC-9. Page 2 Line 13: What does "poorly constrained" mean here? In the context of the sentence, how can the significance of something be poorly constrained?

AR: Mini fractures are a mostly unknown phenomenon (while large-scale fractures and faults have been investigated more frequently), and it is in particular unclear how such pathways for fluid and gas migration will affect the development of methanotrophic communities. We will clarify in the revised MS that such fractures likely change the availability of electron donors and acceptors around the fracture and thus create microbial habitats around the fracture that may differ from sediments at the same sediment depth but away from the fracture

RC-10. Page 2 Line 14: "maintain their metabolism" is technically not a right phrase; better say, ". . .. . . can then be used by sedimentary microbes as metabolic / bioenergetic substrates".

AR: We will changed the wording to..."can be used by microbes as substrates".

RC-11. Page 2 Line 23: "thus" or "therefore" should come before "play".

AR: We will change the wording.

RC-12. Page 2 Lines 27-28: "Using an interdisciplinary approach that combines geochemical and biogeochemical measurements, we investigate the . . ..." – There can be nothing like "biogeochemical measurements"; the whole thing is biogeochemical, so geochemical and biogeochemical are two redundant words; moreover lipid profile

is a microbiological issue, and when combined with isotope ratios the issue becomes geomicrobiological. So this phrase should be written as, "......... geochemical and geomicrobiological methods . . .. . ..."

AR: We will rephrase the wording as suggested.

RC-13. Page 2 Line 37: Must introduce the reader to what "pockmark structures" are; can't pass on the onus of explaining this key concept to other citations.

AR: We will add a brief introduction to pockmark.

RC-14. Page 3 Line 4: 'Enhanced reflections and "push-down" features . . .. . ...' – Please explain these terminologies so as to make the paper more comprehendible and attractive in its professed interdisciplinary context.

AR: Together with the introduction to 'pockmarks', we will clarify these terms.

RC-15. Page 3 Line 6: "High excess" – it's a redundancy of words; just say "Excessive pore pressure".

AR: Yes, we will change the wording here.

RC-16. Page 3 Line 31: ". . .. . ..X-ray analysis in our home laboratories" – I think the authors meant "in the on land laboratory (rather than in the on board or on ship laboratory)"; so it should be written that way. Otherwise it seems that the authors had installed the Geotek MSCL-XCT in their homes.

AR: We will change the text to "onshore laboratories".

RC-17. Page 4 Line 36: "Nevertheless, methane analyses revealed a significant increase in methane concentration along the observed fracture (Fig. 4)." – When you just say "along the observed fracture" and don't mention the trajectory (whether from top to the bottom or upward from the bottom) then the reader gets to construe the first. But actually from top to the bottom of 893 MC, methane is not increasing rather decreasing. So please mention the specific trajectory.

AR: We will change the description of the methane concentration profile to be more comprehensive.

RC-18. Page 7 Lines 3-4: 5. "Assuming a uniform source methane value of XX for the Lomvi pockmark, site 008PC showed the highest $\Delta\delta$13C values (Fig. X)." – major oversight behind this incomplete statement – please rectify (what would be the value of XX?).

AR: We are sorry for this typo that was left from an internal review round. The XX refers to a $\delta$13C source methane value of -55 ‰ which we will correct here and in Figure 5.

RC-19. Table 1 should have a one-line description in relation what is listed in it; for instance, location of coring-stations, seafloor habitat information, analyses performed for individual sample-sites, etc. Incomplete / abbreviated expressions, such as "bac. mats & carb" for crucial habitat-related information, in Table 1, should be avoided. In Table 1 it is appearing that methane concentration and lipid biomarker analysis were not determined in the whole study although that was actually not the case; please amend.

AR: We will change the text here as well as in the caption of table 1 and will edit the table accordingly.

---

## Author Comment (AC2) · 6 Dec 2018

Authors reply to Referee #2:

We thank the anonymous reviewer's positive comment and suggestion. Below we reply to the comment and suggestions one by one.

Referee's comment (RC): However, the following points need to be relooked at

RC-1.Clarity needs to be brought into the naming of the cores, their depth, site distance from each other.

AR: We thank the referee's comment; we will change the text and introduce core location depth and distance to each other more thoroughly in order to clarify the similarities and differences among the investigated cores. We will also give an earlier reference to Tab. 1 and Fig. 1 where these details are shown.

RC-It is unclear when the non-seeping reference site and a high flux Hydrate ridge site is used in the manuscript for comparison.

AR:We compare our findings to a hydrate ridge core (described in Briggs et al., 2011), as this also contains a mini fracture. In fact, Briggs et al., describe the only other sites where such mini fractures were found, thus these are highly relevant for comparison with our study. We will highlight this in more detail in a revised version of the MS.

RC-It is unclear whether only the Lomvi pockmark core has a min fracture or the other cores also had and to what extent. Also when a comparison is done between cores with respect to the fracture zone its important to know how intense was the fracture in all the cores compared and then that can be discussed with respect to the microbial community therein.

AR: Only the Lomvi pockmark core was found to have a fracture. We have stated this already in the original MS (e.g. first line of the abstract) but will edit the result section to highlight that the other cores did not contain a fracture. We found/observed this fracture by X-ray scanning onshore, and because of the gas expansion during core retrieval, it is difficult to estimate how intense the fracture was in situ. We would thus like to refrain from further editing the text in this direction.

RC-2. Please explain the precaution taken during coring so that the reliability of the extent and presence of a mini-fracture is confirmed.

AR: When coring, our aim was initially not to retrieve a core with fractures. E.g., we did not perform autoclave-type coring that would be necessary to retrieve gaseous cores without alterations due to gas expansion. We will mention in the revised version of the MS that such coring techniques would be necessary to better investigate fracture features at quasi in situ conditions. We have mentioned in the text that X-ray analysis can be used to confirm the existence of a mini-fracture in regular multicorer type cores, but the original size and extent of this fracture remain uncertain because of gas expansion during core retrieval.

RC-3. Also there are contradictory findings in this manuscript which needs to be justi-fied appropriately rather than just be assumed.

AR: We are not sure what the reviewer refers to, but presume that our seemingly con-tradicting findings of increasing methane concentrations in core 893MC, which were not paralleled by signals indicative for AOM (such as an increase in alkalinity, decrease in sulfate and increase in biomarker signals indicative for methanotrophic microbes) are meant – see page 5, line 22-29 in the original MS. We attribute this finding to a rather recent increase in methane flux and the opening of the fracture leaving to little time for an AOM community development. In the revised MS, we will further clarify that this is the 'least parsimonious explanation' because under thermodynamic considera-tions, AOM would be favorable in this setting. Thus, the absence of this process and the absence of a well-developed AOM community is best explained by the very slow growth of AOM communities. Indications for alternative factors that could explain this contradiction, e.g. AOM inhibition due to high salt concentrations (see e.g., Steinle et al., 2018) were not found here.

RC-3. Pg 3 Line 31 & Line 40 Repalce the word 'home' with the name of the laboratory

AR: The text will be modified to "the onshore geology laboratory at UiT."

RC-4. Pg 3 ln 36 In the methodology the statement 'Details of the titration protocol can be found in Latour et al. (in review)' is not reader friendly as the paper is under review so it would better to specify the method used.

AR: We will update the text with a brief description of the titration method.

RC-5. Pg4 Ln 13-14 there are three references sited which part of the protocol has

been taken from which reference is not clear to the reader, either it should be given clearly or details should be elaborated in the methodology section.

AR: We will modify the text as follows: Lipid biomarkers were extracted and further analyzed according to a previously reported protocol in Elvert et al. (2003), with modifications for alcohol derivatization (Niemann et al., 2005) and instrumental setup (Blees et al., 2014).

RC-5. Pg 5 Line 5-7 Needs reframing to bring out clarity to the reader

AR: see the previous reply-3.

RC-6. Pg 4 Line 5 What does the word 'highest' mean, it can range from any number, please specify quantitatively

AR: We will change the wording to '. . . that AOM activity at the location of core 008PC was higher than at the other two coring sites.'

RC-7. Pg 4 ln 37. It is not clear how methane concentrations were determined from the fracture zone.

AR: This sentence was a bit misleading and will be changed for more clarity in the revised MS. Methane concentration were measured in a parallel core.

RC-8. Pg 5 ln5-7 Though the contradictory observations are attributed to the recent development of the fracture, it is not clear as to what could be the time period for the word 'recent'.

AR: See above. AOM communities were found to have doubling times of month. Thus the development of an AOM community with 1012 cells per ml sediment or more as is typical for highly active AOM sites, would take years. The absence of such a community at the Lomvi coring site thus indicates that the fracture opened no longer than some years ago. We will include this in the revised text.

RC-9. Pg 5 ln 28 The authors are assuming a process. It would be better if the authors

only explain the possible conditions or mention the factor that could lead to such a function.

AR: In the original MS (page 5, lines 22-25), we have outlined that bioirrigation as well as a recent increase in the methane flux could lead to non steady-state sulfate and methane profiles. Indications for moderate bioirrigation (due to the presence of tube worms) were found and also a recent increase in the methane flux is not unlikely, so that both factors could account for this.

RC-10. pg 6 ln 20 pls specify the location

AR: The text 'the location' will be changed to 'the Lomvi pockmark in Vestnesa Ridge'.

---

## Author Response (AR1)

We are grateful to the two anonymous reviewers' comment and suggestions. In the following file, we have enclosed a point-by-point response to the comments and the revised version of the manuscript with marked-up changes.

Authors reply to Referee #1:

We thank the anonymous reviewer's positive comment and suggestion. Below we reply to the comment and suggestions one by one.

*Referee's comment (RC):*
*1. The authors have used multicorer and push-coring techniques to retrieve the sediment samples. The fractured core (893 MC) in particular was retrieved by the multicorer. While it remains uncertain as to how faithfully a 40 cm core retrieved by this process can preserve small-scale fracture net- works and present them for analysis, the authors too seem to have appreciated the possibility that high internal gas pressure released at the time of the exposure of the core to atmospheric pressure (which is far low than its in situ pressure) may have ex- panded the fracture. The most remarkable fact, also according to the authors, is that despite the above issues, methane significantly increases in concentration as we go up the fracture from the bottom (Fig. 4). My concern is that the text offers no clear explanation regarding why methane concentration is higher near the sediment surface than at the bottom.*

Authors's reply (AR):

We thank the reviewer for suggesting to clarify the concentration differences in the fractured core. We have the text accordingly offering the explanation that the methane concentration is higher near the sediment surface than at the bottom (pg. 5, Line 12-15)

*RC:*

*Whilst a robust explanation of this is absolutely necessary, I would additionally recommend that the lipid biomarker study be complemented with metagenomic and/or metatranscriptomic analyses to reveal the microbial communities more objectively.*

Thanks for the reviewer's recommendation for adding the metagenomic or metatranscriptomic analyses. While the reviewer is right that such analyses would further add to the ID and functioning of the microbial community, we have to admit that this is beyond the author's PhD project, so that we would like to refrain from running further analyses.

*RC-2. For smoother reading and comprehension of the present manuscript, the authors should coin such identifiers (or names) for individual sediment cores that*

*harness both the core numbers as well as the coring-site names.*

We have changed the core names as Lomvi core 893MC, Lomvi core 008PC and Lunde core 886MC so that the location and the core numbers are clear.

*RC-3. In the Methods section, the procedure for measuring methane concentration (the central parameter of this study) must be described and not entrusted entirely on cross referencing.*

We have added the method for methane measurement in pg. 3, line 32-34.

*RC-4. The labels of Figure 2 and 4 lack resolution.*

We have changed the resolution of the labels in Figure 2 and 4.

*RC-5. Page 1 Line 22: I think the readers will find it easier to navigate the data if you say here "In a separate un-fractured core, recovered from the same pockmark approximately 80 m away from the fractured core, we observed . . . . . . . . . . . .".*

We have change the description here as the reviewer suggested.

*RC-6. Page 1 Line 34: "Increased sediment permeability due to fractures may facilitate gas and fluid transport." – I think one should also consider seepage of sulfate, nitrate, etc. from the water into the sediment. Next sentence: no need to start with "Indeed".*

We have removed "indeed". We have mentioned only little about the sulfate into the sediment in some places in the MS for example pg 6, line 1-2. We didn't elaborate on the water column ions to the sediment as this manuscript is to emphasize the "fracture" as channels as the transport mechanism in the sediment whereas the water column sulfate or nitrate are difficult to induce such "fracture".

*RC-7. Page 2 Line 4: Words like "tremendously" are subjective and judgmental so should be avoided.*

We have removed the word "tremendously".

*RC-8. Page 2 Line 5: Instead of obscure words such as "consume methane", or "benefit from ...... sulfide" one should use specific terms as "oxidize".*

We have changed the wording as "oxidize" and "metabolize".

*RC-9. Page 2 Line 13: What does "poorly constrained" mean here? In the context of the sentence, how can the significance of something be poorly constrained?*

We have changed the text as the effects of the mini-fractures are poorly constrained.

*RC-10. Page 2 Line 14: "maintain their metabolism" is technically not a right phrase; better say, ". . .. . . can then be used by sedimentary microbes as metabolic / bioenergetic substrates".*

We have changed the wording to…"can be used by microbes as substrates".

*RC-11. Page 2 Line 23: "thus" or "therefore" should come before "play".*

We have changed to "thus play".

*RC-12. Page 2 Lines 27-28: "Using an interdisciplinary approach that combines geochemical and biogeochemical measurements, we investigate the . . ..." – There can be nothing like "biogeochemical measurements"; the whole thing is biogeochemical, so geochemical and biogeochemical are two redundant words; moreover lipid profile is a microbiological issue, and when combined with isotope ratios the issue becomes geomicrobiological. So this phrase should be written as, "......... geochemical and geomicrobiological methods . . .. . . ..."*

We have changed this to geochemical and organic geochemical methods.

*RC-13. Page 2 Line 37: Must introduce the reader to what "pockmark structures" are; can't pass on the onus of explaining this key concept to other citations.*

We have added the introduction of pockmarks pg 3, line 8.

*RC-14. Page 3 Line 4: 'Enhanced reflections and "push-down" features . . .. . ...' – Please explain these terminologies so as to make the paper more comprehendible and attractive in its professed interdisciplinary context.*

Together with the introduction to 'pockmarks' pg 3, line 8, we clarified these terms line 9-10.

*RC-15. Page 3 Line 6: "High excess" – it's a redundancy of words; just say "Excessive pore pressure".*

We have changed to "excessive pore pressure".

*RC-16. Page 3 Line 31: ". . .. . . ..X-ray analysis in our home laboratories" – I think the authors meant "in the on land laboratory (rather than in the on board or on ship laboratory)"; so it should be written that way. Otherwise it seems that the authors had installed the Geotek MSCL-XCT in their homes.*

We have changed the text to "onshore laboratories at UiT".

*RC-17. Page 4 Line 36: "Nevertheless, methane analyses revealed a significant increase in methane concentration along the observed fracture (Fig. 4)." – When you just say "along the observed fracture" and don't mention the trajectory (whether from top to the bottom or upward from the bottom) then the reader gets to construe the first. But actually from top to the bottom of 893 MC, methane is not increasing rather decreasing. So please mention the specific trajectory.*

We have clarified the description here pg 5, line 11.

*RC-18. Page 7 Lines 3-4: 5. "Assuming a uniform source methane value of XX for the Lomvi pockmark, site 008PC showed the highest Δδ13C values (Fig. X)." – major oversight behind this incomplete statement – please rectify (what would be the value of XX?).*

We have changed the Fig. X to Fig. 5 and XX to be -55.

*RC-19. Table 1 should have a one-line description in relation what is listed in it; for instance, location of coring-stations, seafloor habitat information, analyses performed for individual sample-sites, etc. Incomplete / abbreviated expressions, such as "bac. mats & carb" for crucial habitat-related information, in Table 1, should be avoided. In Table 1 it is appearing that methane concentration and lipid biomarker analysis were not determined in the whole study although that was actually not the case; please amend.*

The description for Table 1 has been changed. And the abbreviations are avoided, the methane and lipid biomarkers analysis are clarify in table 1.

Authors reply to Referee #2:

*Referee's comment (RC):*
*However, the following points need to be relooked at*

*RC-1.Clarity needs to be brought into the naming of the cores, their depth, site distance from each other.*

We have changed the core names as replied earlier to the first referee.

*It is unclear when the non-seeping reference site and a high flux Hydrate ridge site is used in the manuscript for comparison.*

We compare our findings to a hydrate ridge core (described in Briggs et al., 2011), as this also contains the mini fracture. In fact, Briggs et al., describe the only other sites

where such mini fractures were found, thus these are highly relevant for comparison with our study.

We have explained this in pg 6 line 27-28.

*It is unclear whether only the Lomvi pockmark core has a min fracture or the other cores also had and to what extent. Also when a comparison is done between cores with respect to the fracture zone its important to know how intense was the fracture in all the cores compared and then that can be discussed with respect to the microbial community therein.*

Only the Lomvi pockmark core was found to have a fracture. We have stated this already in the original MS (e.g. first line of the abstract) also in the abstract now highlight the other cores are "unfractured" pg 1 line 19. We found/observed this fracture by X-ray scanning onshore, and because of the gas expansion during core retrieval, it is difficult to say how intense the fracture was in situ. We would thus like to refrain from further editing the text in this direction.

*RC-2. Please explain the precaution taken during coring so that the reliability of the extent and presence of a mini-fracture is confirmed.*

When coring, our aim was initially not to retrieve a core with fractures. E.g., we did not perform autoclave-type coring that would be necessary to retrieve gaseous cores without alterations due to gas expansion. We will mention in the revised version of the MS that such coring techniques would be necessary to better investigate fracture features at quasi in situ conditions. We have mentioned in the text that X-ray analysis can be used to confirm the existence of a mini-fracture in regular multicorer type cores, but the original size and extent of this fracture remain uncertain because of gas expansion during core retrieval.

*RC-3. Also there are contradictory findings in this manuscript which needs to be justified appropriately rather than just be assumed.*

We are not sure what the reviewer refers to, but presume that our seemingly contradicting findings of increasing methane concentrations in core 893MC, which were not paralleled by signals indicative for AOM (such as an increase in alkalinity, decrease in sulfate and increase in biomarker signals indicative for methanotrophic microbes) are meant – see pg 5, line 22-29 in the original MS. We attribute this to a rather recent increase in methane flux and the opening of the fracture leaving to little time for an AOM community development.

We have explained these contradictory observations more in the revised MS at pg 5, line 21-25, and pg 6, line 35-42.

*RC-3. Pg 3 Line 31 & Line 40 Repalce the word 'home' with the name of the laboratory*

The text is changed to "the onshore geology laboratory at UiT."

*RC-4. Pg 3 ln 36 In the methodology the statement 'Details of the titration protocol can be found in Latour et al. (in review)' is not reader friendly as the paper is under review so it would better to specify the method used.*

We have updated the text and used the reference from Grasshoff 1999.

*RC-5. Pg4 Ln 13-14 there are three references sited which part of the protocol has been taken from which reference is not clear to the reader, either it should be given clearly or details should be elaborated in the methodology section.*

We modified the text as follows: Lipid biomarkers were extracted and analyzed according to previously reported protocols (Elvert et al., 2003) with modification for alcohol derivatization (Niemann et al., 2005), and instrument set up (Blees et al., 2014;Steinle et al., 2018).

*RC-5. Pg 5 Line 5-7 Needs reframing to bring out clarity to the reader*

see the previous reply-3.

*RC-6. Pg 4 Line 5 What does the word 'highest' mean, it can range from any number, please specify quantitatively*

We have changed the wording here to '… that AOM activity at the location of core 008PC was higher than at the other two coring sites.'

*RC-7. Pg 4 ln 37. It is not clear how methane concentrations were determined from the fracture zone.*

We have clarified in the text that "The methane headspace samples were obtained on a parallel core as the fractured core in the same set of multicorer frame."

*RC-8. Pg 5 ln5-7 Though the contradictory observations are attributed to the recent development of the fracture, it is not clear as to what could be the time period for the word 'recent'.*

In pg6 line 38-40, we mentioned the time to be "a few years".

*RC-9. Pg 5 ln 28 The authors are assuming a process. It would be better if the*

*authors only explain the possible conditions or mention the factor that could lead to such a function.*

In the original MS (pg5, lines 22-25), we have outlined that bioirrigation as well as a recent increase in the methane flux could lead to non steady-state sulfate and methane profiles. Indications for moderate bioirrigation (due to the presence of tube worms) were found and also a recent increase in the methane flux is not unlikely, so that both factors could account for this. In the revised version pg 5, line 40 and pg 6, ling 1-5.

*RC-10. pg 6 ln 20 pls specify the location*

The text 'the location' is changed to 'the Lomvi pockmark in Vestnesa Ridge'.